# MIGOU: A Low-Power Experimental Platform with Programmable Logic Resources and Software-Defined Radio Capabilities

**DOI:** 10.3390/s19224983

**Published:** 2019-11-15

**Authors:** Ramiro Utrilla, Roberto Rodriguez-Zurrunero, Jose Martin, Alba Rozas, Alvaro Araujo

**Affiliations:** B105 Electronic Systems Lab, ETSI Telecomunicación, Universidad Politécnica de Madrid Av. Complutense 30, 28040 Madrid, Spainjose.martin.burgos@gmail.com (J.M.); albarc@b105.upm.es (A.R.); araujo@b105.upm.es (A.A.)

**Keywords:** platform, IoT, end-device, cognitive radio, edge computing, software-defined radio

## Abstract

The increase in the number of mobile and Internet of Things (IoT) devices, along with the demands of new applications and services, represents an important challenge in terms of spectral coexistence. As a result, these devices are now expected to make an efficient and dynamic use of the spectrum, and to provide processed information instead of simple raw sensor measurements. These communication and processing requirements have direct implications on the architecture of the systems. In this work, we present MIGOU, a wireless experimental platform that has been designed to address these challenges from the perspective of resource-constrained devices, such as wireless sensor nodes or IoT end-devices. At the radio level, the platform can operate both as a software-defined radio and as a traditional highly integrated radio transceiver, which demands less node resources. For the processing tasks, it relies on a system-on-a-chip that integrates an ARM Cortex-M3 processor, and a flash-based FPGA fabric, where high-speed processing tasks can be offloaded. The power consumption of the platform has been measured in the different modes of operation. In addition, these hardware features and power measurements have been compared with those of other representative platforms. The results obtained confirm that a state-of-the-art tradeoff between hardware flexibility and energy efficiency has been achieved. These characteristics will allow for the development of appropriate solutions to current end-devices’ challenges and to test them in real scenarios.

## 1. Introduction

The number of mobile and Internet of Things (IoT) devices continues to grow at an unprecedented rate [1]. In addition, these systems increasingly have better performance, so new and more demanding applications and services are continuously appearing. These growth trends entail a substantial increase in wireless data traffic, and therefore, spectrum occupancy, which in turn poses new and important challenges to overcome. One of these challenges is to achieve highly efficient and reliable communications in increasingly complex, heterogeneous, and dynamic scenarios where many different radio systems coexist. To this end, multiple cognitive radio (CR) techniques are being proposed to allow the devices to sense and analyze the spectrum, and based on that information and on their previous experience, make decisions about its use and adapt their communication parameters accordingly [2,3].

On the other hand, applications and services increasingly demand a more comprehensive understanding of the environment, i.e., a higher level of information is sought through the processing of raw sensor data [4,5]. Given the spectrum scarcity problem, it is not appropriate to perform this processing in the cloud since it would generate latency problems and massive network traffic due to the data uploading. For this reason, and due to the improvements in the end-devices features and performance, edge computing is gaining importance in areas like the IoT [6,7]. This paradigm argues that computing and storage capabilities must be placed where the data is generated, i.e., in the end-devices, which must be responsible for all or at least some of this processing. This approach allows for faster response times and reduces communication overheads, which in turn means energy savings.

However, important CR and processing tasks have a strong dependence on the hardware architecture and resources of the devices. For example, most of the CR techniques proposed in the literature are based on software-defined radio (SDR) platforms since they allow for direct access to the in-phase and quadrature (I/Q) samples of the baseband signals and have a great radio flexibility. These features offer many more possibilities to sense and adapt to the radio environment than those of commercial off-the-shelf (COTS) transceivers, whose rigid application-specific integrated circuit (ASIC) implementation severely limits experimentation with the lower layers of the communication stack [8]. Nevertheless, current SDR platforms have not been designed from the perspective of battery-powered end-devices, so they rely on high-performance chips, whose power consumption and price are very high [9]. In addition, since the complexity and latency restrictions of processing tasks are continuously increasing, the use of hardware solutions for their implementation, such as FPGAs, becomes more common [10]. However, in the same way, they usually rely on high-performance development platforms [11], with the same aforementioned drawbacks, or on custom solutions for specific applications [12,13].

Current end-devices, such as TelosB [14], MICAz [15], or YetiMote [16], do not usually have those hardware processing resources and are equipped with COTS transceivers, thus achieving a very low power consumption. This permits experimentation in real scenarios with long-term deployments formed using numerous units, which in turn allows developers to evaluate how their protocols or techniques scale. However, this system architecture limits CR and edge computing research in areas of great interest, such as IoT, wireless sensor networks (WSNs), or mobile sensing, because of the significant architectural constraints it presents.

Consequently, this work aims to address the lack of an appropriate platform for these research areas. Specifically, our contribution is a wireless experimental platform that simultaneously addresses the energy-efficiency requirements of end-devices and the hardware flexibility demanded by the current CR and edge computing challenges. The presented solution can operate both as an SDR and as a highly integrated radio transceiver, which demands less node resources. Besides, it relies on a system-on-a-chip (SoC) that integrates an ARM Cortex-M3 processor, and a flash-based FPGA fabric, where high-speed processing tasks can be offloaded. The platform supports YetiOS [17], an operating system (OS) that speeds up the development of applications and tests. The power consumption of the platform was measured in the different modes of operation. The results obtained confirm that a state-of-the-art tradeoff between hardware flexibility and energy efficiency was achieved.

The rest of the paper is organized as follows. Section 2 presents related works in experimental platforms for CR and edge computing. In Section 3, we introduce our platform, detailing the design decisions and its hardware architecture. Section 4 contains a full description of the resources and methodology used in the platform evaluation. The results obtained in the different tests are presented and discussed in Section 5. Conclusions are offered in Section 6.

## 2. Related Work

Nowadays, there are many SDR platforms that are widely used both in academia and industry. Among them, the most broadly used are Ettus Research USRPs [18], BladeRF [19], HackRF [20], WARP [21], ADALM-PLUTO [22], and LimeSDR [23]. However, as mentioned in the previous section, all of them are based on high-performance chips, so their power consumption is in the order of watts, something unsuitable in most cases for a battery-powered end-device. Moreover, all these platforms are only radio frontends, so an additional processor-based system is required to manage them.

Multiple works [24,25,26] use the RTL-SDR USB dongle [27], which is a low-cost commercial digital video broadcasting terrestrial (DVB-T) TV receiver, along with a Raspberry Pi as an SDR platform. The power consumption of this solution is still above one watt, but it is a very economical option, and thanks to the use of the Raspberry Pi, allows for the use of helpful SDR libraries, such as GNU Radio [28]. Apart from consumption, another drawback of this solution is that it only embeds a receiver, which considerably limits the development options.

MarmotE SDR [29] and µSDR [30] are two low-cost and sub-watt SDR platforms, which were specifically designed to be battery-powered. In terms of hardware architecture, both are practically equivalent. Their processing backend is based on the Microsemi SmartFusion SoC. This device integrates a flash-based FPGA fabric and an ARM Cortex-M3 processor. Thus, it gives more flexibility than traditional fixed-function microcontroller units (MCUs) without the excessive cost of soft processor cores on traditional FPGAs. Moreover, compared to SRAM-based FPGAs, flash-based FPGAs have a much lower static consumption, which is crucial for low-power operation. As a radio frontend, both platforms use a 2.4 GHz I/Q transceiver of the MAX283X family from Maxim Integrated. This device allows for direct access to the I/Q components of the baseband signals and it integrates almost all the circuitry required to implement the radio frontend functions.

These two works have proved the feasibility of this kind of platform in terms of cost, size, power consumption, and the utilization of FPGA logic resources to approach the CR research from the end-device perspective. However, it is still possible to go further. When analyzing the consumption data of MarmotE SDR [8,29], it can be seen that the radio transceiver is responsible for more than half of the power consumption in those modes in which it is active, reaching 60% of the total consumption in the receive mode and 86% in the transmit mode. Therefore, this component constitutes one of the most critical parts of the design, with a high impact in the overall energy utilization. In addition, it only operates in the 2.4 GHz band, limiting the possibilities of experimentation. On the other hand, in sleep mode, MarmotE SDR has a total power leakage of 70.8 mW, of which 50 mW (71%) are drawn by the SmartFusion and 20 mW (28%) by the temperature compensated crystal oscillator (TCXO), which is used to drive the radio transceiver and is always enabled by design, even in this low-power mode. Moreover, the SmartFusion does not support any ultra-low-power mode, while its new version, SmartFusion2, does.

As shown in the results of this work, the consumption of these low-power SDR platforms is still much larger than that of traditional end-devices. In addition, the implementation and evaluation of the PHY/MAC layers for SDR systems implies a major commitment of time and money because of its complexity. This is a price that must be paid when working with SDRs, even when these layers do not include any cognitive feature. The lack of a platform that allows for benefitting from both the flexibility and processing capacity of SDRs, and the energy efficiency and agility of development of traditional nodes, has inspired this work.

## 3. MIGOU Platform

### 3.1. Design Criteria

The design of our platform has been based on the following criteria:*Hybrid radio*: The platform must be able to operate as a traditional node, using a highly integrated transceiver, and as an SDR system when a high radio flexibility is required for specific CR tasks. It should be noted that when no special task or customization is required, the use of an exclusively SDR system results in a considerable penalty in terms of power consumption, resource usage, and development time. This idea is further detailed in a previous work [31].*Low-power programmable logic*: The platform must include programmable logic resources where high-speed communication and/or processing tasks can be offloaded. This is especially important for applications with strict timing constraints, where it has been shown that these tasks can affect each other negatively [32]. Besides, these logical resources must support some low-power consumption mode for when they are not in use.*Flexible hardware–software (HW–SW) boundary*: The communication interface between the MCU, where the application runs, and the programmable logic must be as fast as possible to avoid bottlenecks, thus ensuring a flexible HW–SW boundary. This is an important feature for communication and processing tasks with precise timing requirements.*Agile application development*: Being an experimentation-oriented platform, the development of new applications should be as agile as possible. In this sense, the use of an OS is considered an essential aspect of the platform. In addition, it should have development and expansion interfaces that maximize its possibilities for use.*Consumption measurement*: The platform should integrate power consumption metering tools that allow the developer to evaluate the energy performance of the methods and algorithms being tested on it.

### 3.2. Design and Implementation

The hardware architecture of the MIGOU platform is presented in Figure 1. This diagram shows the main electronic components that constitute the system, as well as their connections. Both the architecture design and the selection of components were carried out according to the previously presented design criteria.

Atmel AT86RF215 was the component selected to provide the platform with hybrid radio capabilities. As shown in Figure 2, this device is comprised of two highly integrated radio transceivers, which operate in the sub-1 GHz and 2.4 GHz bands. Moreover, each of these transceivers supports two operating modes, which do not need to be the same. The ability to switch between these two modes of operation is what allows our platform to operate as a traditional node or as an SDR system.

On the one hand, in baseband (BB) mode, the AT86RF215 works as a COTS transceiver. In this configuration, the selected transceiver is internally paired with an optimized baseband core (BBC) that supports a wide variety of data rates with three modulation schemes. This allows for a highly efficient implementation of multiple PHY layers. In this mode, the BBC controls the radio and processes the data encoding/decoding for transmitting (TX) and receiving (RX) from the internal frame buffers. On the other hand, in the I/Q mode, the radio is controlled from an MCU via a serial peripheral interface (SPI) and the I/Q data stream is routed directly between the selected radio and a serial low voltage differential signal (LVDS) interface. This configuration allows for the use of a baseband processor for implementing custom SDR features.

We chose the SmartFusion2 as a control unit and mixed-signal processing backend. This device integrates a microcontroller subsystem (MSS), based on an ARM Cortex-M3 processor, and a flash-based FPGA. This kind of FPGA has been proved to have faster wake-up times and to be more energy-efficient than SRAM-based FPGAs [29]. In addition, the SmartFusion2 family, unlike the SmartFusion, supports a low-power mode of the FPGA fabric, which is known as the Flash*Freeze mode. This is the technology selected to provide our platform with low-power programmable logic resources. Furthermore, the ARM Cortex-M3 and the FPGA communicate internally through an AMBA high-performance bus (AHB). This bus is widely used to connect peripherals, memories, and cores in SoCs, and it provides a bandwidth of up to 16 Gbps, minimizing potential data bottleneck problems, and therefore, guaranteeing a flexible HW–SW boundary.

Therefore, most of the required elements of our platform are mainly found in just two chips that can be connected directly, namely the AT86RF215 and the SmartFusion2. Specifically, as shown in Figure 3, when the platform is in I/Q mode, the ARM Cortex-M3 is responsible for setting and controlling the AT86RF215 via an SPI, and the I/Q samples are transferred through the LVDS interface between the selected radio and the FPGA, where they are processed. On the other hand, when the platform is in BB mode, the ARM Cortex-M3 controls the AT86RF215 via an SPI, and the FPGA can be put in Flash*Freeze mode if no other high-speed processing task is required.

Although the platform could already operate autonomously with these two chips, the possibility of using it in conjunction with a computer (PC) during the process of developing and evaluating new techniques is very convenient, since it allows for the use of high-level development and debugging tools. For this reason, the platform has two communication interfaces. On the one hand, the FTDI FT232R UART–USB converter allows the developer to control and receive information from it through a terminal on the PC. On the other hand, the platform also has a synchronous parallel to USB 2.0 hi-speed interface based on the FTDI FT232H chip. This allows the I/Q data stream to be sent to or received from the computer. This chip supports transfers of up to 320 Mbps, which is more than enough considering that the LVDS interface between the AT86RF215 and the SmartFusion2 has a maximum data rate of 128 Mbps. Furthermore, it should be mentioned that since these interfaces are only used for developmental tasks, both are powered from the USB port, so they do not affect the platform power consumption. Moreover, FTDI, the manufacturer of these two chips, provides drivers and application programming interfaces (APIs) for handling them both from Linux and Windows OSs. Apart from these two specific interfaces, the MIGOU platform has two 24-pin expansion connectors with general-purpose inputs/outputs (GPIOs), UART, SPI, and I2C interfaces, and it also includes a temperature sensor (Texas Instruments AT30TS75A) and an accelerometer (NXP MMA8652FC) on the board itself.

The platform can be powered from the two mentioned USB ports by using a power adapter or a single cell Li-ion battery. In addition, MIGOU includes the Texas Instruments BQ27441 fuel gauge to monitor battery parameters, such as the remaining capacity, state-of-charge, voltage, or state-of-health. The management of the different power sources and the battery charge is done through the BQ24166 power management integrated circuit (PMIC) by Texas Instruments. This device provides a stable output voltage of 4.2 V, which in turn is used as the input of the DC–DC converters that generate the necessary voltages for the platform. Specifically, two Texas Instruments TPS62097 converters are used to generate the lower voltages of 1.2 V and 1.8 V, and two Texas Instruments TPS62150A-Q1 converters generate 2.5 V and 3.3 V. All these converters were selected for their high efficiency and the small number of external components they require. The platform includes a selector of the supply voltage of the transceiver (V_TRX_ SEL), so its performance and power consumption can be evaluated at different voltage levels.

Since one of the purposes of the platform is to evaluate the energy efficiency of new methods and solutions, MIGOU includes the STMicroelectronics STM32L496RGT6, an auxiliary MCU (AUX MCU) exclusively dedicated to monitor its current consumption. Specifically, as shown in Figure 1, it monitors the power supply line of the entire system, V_SUP_, and that corresponding to the radio, V_TRX_. For each line, the MCU switches a pair of sensing resistors, which allows it to measure in a range from 1 µA to 100 mA with V_TRX_ and from 300 µA to 300 mA with V_SUP_. With these two monitoring points, the total consumption of the platform and the partial consumptions corresponding to each of the two main parts, the frontend and the backend, can be known. In order to not alter these measurements, the auxiliary MCU has an independent power supply. The current consumption measurements can be sent to an external computer through a USB port, or they can be stored in a MicroSD memory card, which allows for remote monitoring over long periods of time.

Figure 4 shows the MIGOU platform and the physical location of its main parts. The printed circuit board (PCB) has four layers and its size is 148 × 86 mm. The platform schematics, bill of materials, and manufacturing files are open access and available online (see the Appendix A at the end of this document).

Finally, at a software level, the MIGOU platform implements YetiOS [17], an OS that speeds up the development of applications and tests. YetiOS is built on top of FreeRTOS [33] and provides several extra features, such as standard input–output (STDIO), a command shell, and GPIOs. It also has advanced memory, timing, and process management modules; Linux-like device drivers; and an adaptive core that allows the OS to improve its performance in dynamic environments. In addition, YetiOS maintains the same preemptive round-robin priority scheduler as FreeRTOS, and it includes a layered network stack, in which each layer is implemented independently, providing a useful framework for designing new protocols. As part of this work, the specific drivers for our platform have also been implemented and integrated within YetiOS.

## 4. Materials and Methods

### 4.1. Materials

MIGOU has been evaluated and compared with other representative platforms in terms of power consumption and hardware features. Specifically, these platforms are:*YetiMote* [16]: This platform was chosen as a representative traditional wireless sensor node. Its architecture is based on a STMicroelectronics STM32L476RE microcontroller and three COTS radio transceivers, two STMicroelectronics SPIRIT1 operating in the 433 MHz and 868 MHz bands, and a Texas Instruments CC2500 for the 2.4 GHz band.*MarmotE SDR* [29]: This platform was selected as a representative low-power SDR system, which was specifically designed to be battery-powered. In addition, amongst the two presented in Section 2, this is the one with more detailed power consumption data. As mentioned before, it is mainly based on a Maxim Integrated MAX2830 I/Q transceiver and a Microsemi SmartFusion SoC.*B200mini and B210 USRPs* [18]: Within the group that we consider high-performance SDR platforms, these two were chosen for having lower features and being widely used. The B200mini model is based on a Xilinx Spartan-6 XC6SLX75 FPGA and an Analog Devices AD9364 radio transceiver. On the other hand, the B210 model, which is more powerful, consists of a Xilinx Spartan-6 XC6SLX150 FPGA and an Analog Devices AD9361.

As for the proposed MIGOU platform, it should be mentioned that there are multiple models of the main SmartFusion2 SoC, each with a different amount of logical and memory resources. Our platform design is pin compatible with many of them for flexibility and scalability. The MIGOU platforms used in this work have been manufactured with the model M2S050, equipped with 56,340 logic elements, 256 kB of embedded non-volatile memory (eNVM), and multiple SRAM blocks distributed between the MSS and the FPGA.

The MIGOU power consumption measurements presented in the results of this work were taken with the Keysight B2902A precision source-measurement unit (SMU) [34]. This instrument was also used to calibrate the dedicated current-measuring hardware that our platform incorporates and to measure the YetiMote power consumption in sleep mode, where high precision is required due to its very low power consumption. On the other hand, the Ruideng AT35 USB Tester [35] was used to measure the power consumption of the USRPs and the YetiMote operating in its active modes. Spectrum analyzers were used to calibrate the transmission power of all of them. Finally, the power consumption data of the MarmotE SDR platform used in this work was not directly measured by us. Instead, we used the empirical data published by its authors [8,29]. 

### 4.2. Methods

The power consumption of the platforms was evaluated in transmit, receive, and sleep modes. To that end, the different platforms were set to statically operate in these modes. The supply voltage and current consumption were measured in each case in order to obtain the power consumption data. The B2902A SMU has a supply voltage resolution of 10 µV, and it measures the current consumption with a resolution of 100 nA. For our tests, this SMU was configured with a sampling rate of 1 kHz, and 2000 samples (window of 2 s), which were averaged to obtain each final measurement. In the case of MIGOU, its dynamic consumption (as a function of time) was also evaluated. In this case, only raw current measurements were used. On the other hand, the Ruideng AT35 provides voltage and current values with a resolution of 1 mV and 100 µA, respectively, and it has a measurement refresh rate of 0.5 Hz. Therefore, in this case each result corresponded to a single measurement (window of 2 s).

The experiments with which the MarmotE SDR platform was characterized [8,29] were used as a reference to be able to perform a comparison between the different systems under similar conditions. Specifically, they implemented a PHY layer in the FPGA that is controlled from the MSS. This layer uses a Gaussian minimum-shift keying (GMSK) modulation with a bandwidth–time (BT) product BT = 0.5. It operates in the 2.4 GHz band with a transmission data rate of 250 kbps and an output power of 0 dBm. During the transmit and receive modes, both the FPGA and the MSS were running at 10 MHz. Moreover, they define the MarmotE SDR sleep mode with the MAX2830 transceiver in shutdown mode; the FPGA in reset mode; the required MSS peripherals running at 32 kHz; and the Cortex-M3 halted, waiting for interrupt (WFI). In their design, the TCXO is always enabled, even in this mode. With this configuration they obtained the lowest-power mode from which the system could wake up.

On the other hand, as mentioned before, MIGOU supports two modes of operation: BB mode, in which it behaves like a traditional node, and I/Q mode, in which it operates as an SDR system. As shown in Figure 3, depending on the selected mode, different subsystems of the platform were enabled or disabled to ensure an energy-efficient operation. In these active modes, the SmartFusion2 was running at 50 MHz and the AT86RF215 at 26 MHz. In addition, a sleep mode is defined in which the AT86RF215 transceiver is set to deep sleep, where the TCXO is disabled; the FPGA is set to Flash*Freeze mode; the Cortex-M3 is halted and put in WFI; and the necessary peripherals of the MSS remain running at 1 MHz, which is the minimum standby clock frequency currently supported by Microsemi. The supply voltage of the transceiver in all the tests was 1.8 V. Table 1 shows a summary of the status of each of the main MIGOU subsystems in the different modes of operation.

To evaluate the power consumption of MIGOU in I/Q mode, the same PHY layer and configuration parameters as in the case of MarmotE SDR were used. Alternatively, to evaluate its power consumption in BB mode, an application was developed in YetiOS to continuously send 64-byte packets from a transmitting node to a receiving one. This application waits 5 ms after a packet has been transmitted to send the next one. In this case, the AT86RF215 was configured to operate in the 2.4 GHz band, but it used a binary frequency-shift keying (BFSK) modulation since the GMSK used in the MarmotE SDR experiments was not supported. In both modulations, the transmitter switches between two alternative frequencies according to the data symbols. The main difference between these two modulations is that in the case of GMSK, this transition between frequencies is carried out in a controlled manner. Furthermore, the AT86RF215 in this configuration does not allow for a data rate of 250 kbps, so we used 200 kbps, which was the closest allowed value.

The same application developed to evaluate MIGOU in BB mode was also used to evaluate the YetiMotes since they also implement YetiOS. In their active modes, YetiMotes run at 48 MHz. In these cases, the radio interfaces operating in the sub-1 GHz bands are disabled and the CC2500 transceiver, which operates at 2.4 GHz, is configured to use a GMSK modulation, a data rate of 250 kbps, and a transmission power of 0 dBm. In the sleep mode, the three transceivers have their crystal oscillators switched off, so they all are in their lowest-power consumption mode. In addition, the MCU is put in *Stop 2* mode, having the main oscillators and most peripherals switched off (Cortex-M4 halted and WFI), while the low-power 32 kHz oscillator keeps running. 

Finally, in the case of the two USRPs, GNU Radio was used to develop a GMSK transmitter and receiver. The transmitter operates with a data rate of 250 kbps and an output power of 0 dBm. Based on the sampling rate required by these systems, both USRPs automatically configure their internal clock to 32 MHz. The USRPs do not support any low-power mode, so their reset status (non-programmed) was considered to be their sleep mode since this is when they consume the least power. The consumption measurements of the USRPs did not include the consumption of the processor-based system that was additionally required for their operation. Table 2 shows a summary of the configuration parameters used in each platform to perform its evaluation.

## 5. Results and Discussion

The hardware features of the different platforms evaluated are compared in Table 3. In addition, their power consumption in the different modes of operation is represented in Figure 5.

First, it should be noted that USRPs lack a processor, even when they require one to be used. This is because these platforms have been conceived as peripherals and not as autonomous devices. As explained in the introduction, they rely on high performance chips; therefore, as shown in Figure 5, their consumption is much higher. However, they also have a better radio performance than the other platforms with a wider operating frequency range (70 MHz to 6 GHz), higher instantaneous bandwidth (56 MHz), etc.

On the other hand, YetiMote is the most energy-efficient platform. It consumes 103.9 mW in transmit mode, 96.3 mW in receive mode, and only 17 µW in sleep mode. This efficiency is possible thanks to the use of highly integrated COTS transceivers and an ultra-low-power MCU. However, this architecture, with a single processor and without programmable logic where certain tasks can be parallelized, can be a limitation for those applications with a high computational load or with strict time requirements. In terms of radio flexibility, this platform has three transceivers that allow it to operate in three different bands, but none of them provide access to the baseband I/Q signals; therefore, experimentation with the lower layers of the communications stack is constrained to the transceivers’ configuration options. In conclusion, YetiMote is a suitable platform for long-term deployments thanks to its low energy consumption. However, it presents important limitations for experimentation and research in CR and edge computing.

In between of these two kinds of representative platforms are MarmotE SDR and our proposed platform MIGOU. The former consumes 287.4 mW in receive mode and 851.7 mW when transmitting. As already mentioned in Section 2 and as can be seen in Figure 5, the MAX2830 transceiver represents 60% and 86% of those power consumptions, respectively. If we compare these values with those of the MIGOU platform operating in I/Q mode, like an SDR system, a large improvement is observed, especially in transmit mode. This reduction is mainly due to the new radio front-end used, which went from consuming 733.7 mW in transmission mode to 95.4 mW, and from 172.4 mW in reception mode to 55.9 mW. This change has many implications in terms of performance. First, the maximum I/Q sample rate went from 22 MSps in MarmotE SDR to 4 MSps in MIGOU, which limits the communication bandwidth and the instantaneous sensing. On the other hand, while the MAX2830 transceiver only operates in the 2.4 GHz band, the AT86RF215 also supports the operation of multiple sub-1 GHz bands, covering the ranges from 389.5 to 510 MHz and from 779 to 1020 MHz. Finally, the baseband cores included in the AT86RF215 transceiver represent a highly differentiating feature between both solutions. They allow for a highly efficient implementation of multiple PHY layers, which gives MIGOU the hybrid radio capability, i.e., being able to operate as a traditional node when the flexibility of the SDR is not necessary. In this way, the development time, the computational load of the node, and the power consumption were reduced, as seen in Figure 5.

Specifically, in transmit mode, the power consumption is reduced by 45.5%, from 299.6 mW in I/Q mode to 163.4 mW in BB mode. More precisely, the consumption of the AT86RF215 drops from 95.4 mW to 49.3 mW, and that of the SmartFusion2 from 200.9 mW to 110.8 mW, as a result of putting the FPGA in Flash*Freeze mode. In receive mode, consumption is reduced by 23.3%, down from 55.9 mW to 43.0 mW in the transceiver and from 142.2 mW to 108.1 mW in the SoC. It should be noted that in both transmit and receive modes, the power consumption of MIGOU’s SmartFusion2 is higher than that of the SmartFusion on MarmotE SDR. This is because the first one was running at 50 MHz, while the second one was running at 10 MHz.

In sleep mode, MIGOU consumes 22.8 mW, 32.2% of the MarmotE SDR’s consumption in the same mode. This is mainly due to two factors. First, because in MarmotE SDR, the TCXO remains active during this mode, which is solvable with minimal changes. The other factor is the Flash*Freeze mode offered by the SmartFusion2. While the SmartFusion consumes 50 mW, the SmartFusion2 only consumes 22.8 mW. There is still a large difference in the power consumption in the sleep mode between MIGOU and the YetiMote-type platforms. This is due to the different performance of the low-power modes of the MCUs and of this type of SoC. However, it is still possible to reduce the power consumption of MIGOU by using an SmartFusion2 with a smaller amount of logical elements, which would limit its processing capacity, or to use an MCU and an independent Microsemi Igloo2 FPGA [13]. In this way, it would be possible to cut the power of the FPGA externally during the sleep mode. However, this approach compromises response times, and the throughput and communication flexibility between the two devices. For these reasons, and since the objective of this work is not only to develop a low-power experimental platform, but also one with a high flexibility and high performance, the benefits of the SoC-based approach were finally considered more appropriate.

As mentioned in Section 4.2, each MIGOU power consumption value presented so far corresponds to the average of 2000 samples taken in a 2-s window. These data, as well as the maximum and minimum values in that window, are shown in Table 4. In addition, Figure 6 shows the raw power consumption measurements of our platform as a function of time. Observing both the table and the figure, two aspects stand out: power consumption in BB mode has a larger range of oscillation, and it changes in a faster and more repetitive way. This is because in this mode, the communication between the AT86RF215 and the SmartFusion2 is based on packets and, both when transmitting and receiving, the transceiver switches between different states of a finite state machine, with each state having a different power consumption. By contrast, in I/Q mode, the communication is based on a constant I/Q samples stream, and the transceiver stays in a fixed state.

Finally, to demonstrate MIGOU’s ability to operate as a hybrid radio system, an application that dynamically switches between its different modes of operation was developed. The sequence in which the modes change is based on a typical CR application in which a node wakes up from sleep mode, senses the spectrum (RX I/Q mode) to adapt its communication parameters to the environment, transmits the corresponding information (TX BB mode), and returns to sleep mode. Figure 7 shows the power consumption of MIGOU while running this application. In addition, the mode of operation in which the platform is at each moment is also shown at the top.

Although the application was idle during the sleep mode, some power consumption peaks were observed during that time. They were caused by the OS management tasks that sporadically woke up the ARM Cortex-M3 processor.

## 6. Conclusions

In this work, we present MIGOU, a low-power wireless experimental platform that was designed to simultaneously address the energy-efficiency requirements of resource-constrained end-devices and the hardware flexibility demanded by the current CR and edge computing paradigms. This platform relies on the SmartFusion2 SoC that integrates an ARM Cortex-M3 processor and a flash-based FPGA, where high-speed processing tasks can be offloaded and computed more efficiently via hardware acceleration. In addition, at the radio level, the platform can operate both as a traditional node, which demands lower energy resources and development time, and as an SDR system, which allows for the implementation of custom cognitive radio features. Moreover, the ability to dynamically switch between these two modes of operation opens the possibility for developing new hybrid strategies, taking advantage of both the flexibility offered by the SDR and the efficiency of the transceiver’s highly optimized baseband cores.

The power consumption of our platform was measured in transmit, receive, and sleep modes. These measurements were compared with the corresponding ones of other representative tools and systems: YetiMote, a traditional IoT end-device; MarmotE SDR, a low-power SDR system; and B200mini and B210 USRPs, two widely used high-performance SDR platforms. Moreover, all these devices were compared in terms of their hardware features. The results obtained confirmed that a state-of-the-art tradeoff between hardware flexibility and energy efficiency was achieved. These features will allow researchers to develop appropriate solutions to current end-devices’ challenges, and to test and evaluate them in real scenarios.

## Figures and Tables

**Figure 1 sensors-19-04983-f001:**
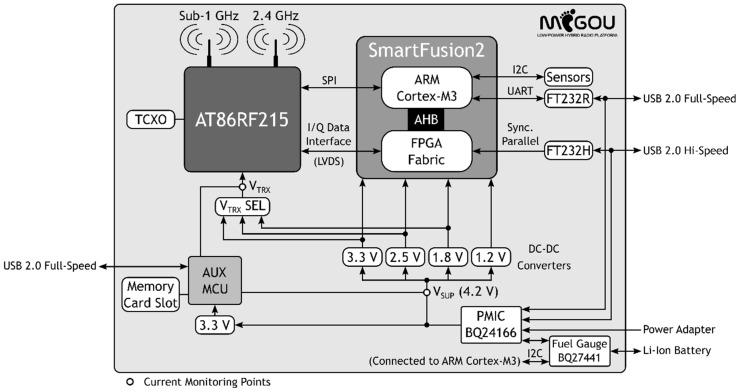
MIGOU platform hardware architecture.

**Figure 2 sensors-19-04983-f002:**
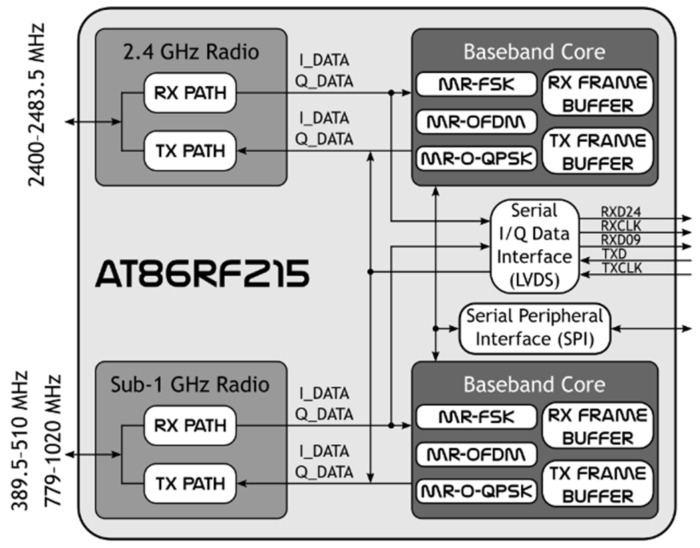
AT86RF215 block diagram.

**Figure 3 sensors-19-04983-f003:**
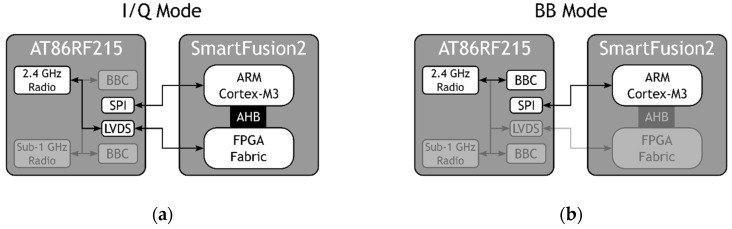
Active subsystems of the MIGOU platform in I/Q mode (**a**) and BB mode (**b**).

**Figure 4 sensors-19-04983-f004:**
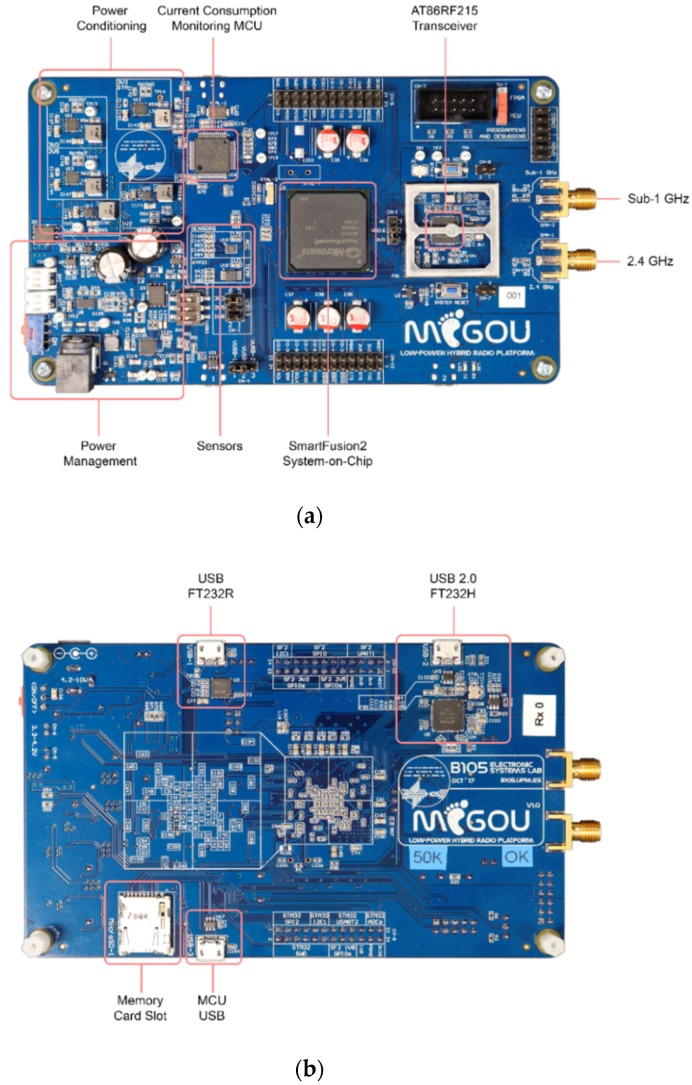
Top (**a**) and bottom (**b**) views of the MIGOU platform.

**Figure 5 sensors-19-04983-f005:**
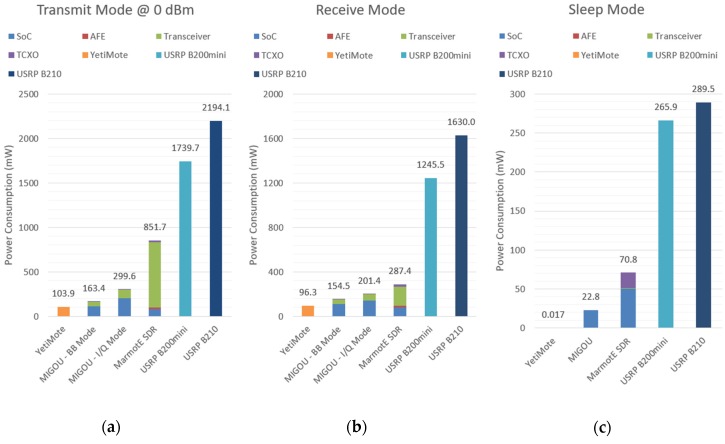
Power consumption of the evaluated platforms in the following modes of operation: (**a**) transmit mode, (**b**) receive mode, and (**c**) sleep mode.

**Figure 6 sensors-19-04983-f006:**
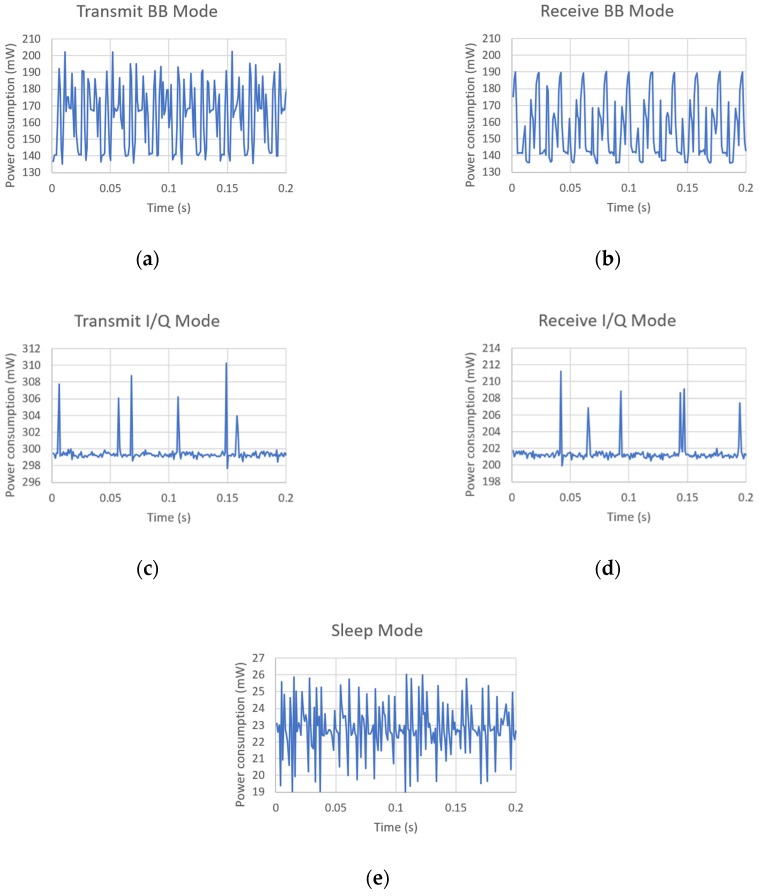
Power consumption of MIGOU as a function of time in the evaluated modes: (**a**) transmit in BB mode, (**b**) receive in BB mode, (**c**) transmit in I/Q mode, (**d**) receive in I/Q mode, and (**e**) sleep.

**Figure 7 sensors-19-04983-f007:**
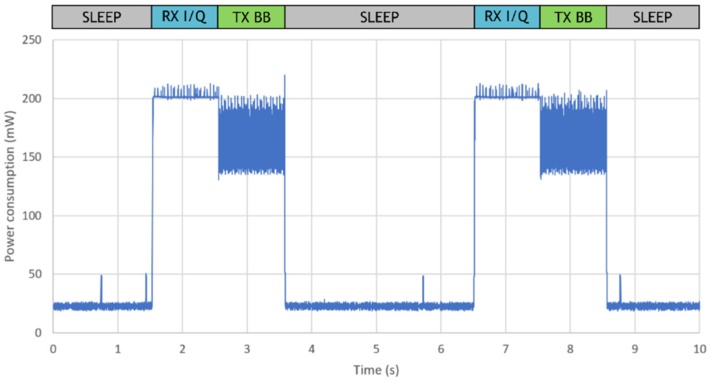
Power consumption of MIGOU in a hybrid radio application that dynamically changed the modes of operation.

**Table 1 sensors-19-04983-t001:** Status of the main subsystems of MIGOU in the different modes of operation.

Chip	Subsystem	I/Q Mode	BB Mode	Sleep
SmartFusion2 (M2S050)	FPGA	ON @ 50 MHz	Flash*Freeze	Flash*Freeze
	MSS	ON @ 50 MHz	ON @ 50 MHz	WFI @ 1 MHz ^1^
AT86RF215 (V_TRX_ = 1.8 V)	Sub-1 GHz Radio	OFF	OFF	OFF
	Sub-1 GHz BBC	OFF	OFF	OFF
	2.4 GHz Radio	ON	ON	OFF
	2.4 GHz BBC	OFF	ON	OFF
TCXO	-	ON	ON	OFF

^1^ The 32 kHz standby clock is no longer supported by Microsemi.

**Table 2 sensors-19-04983-t002:** Configuration parameters used in the evaluation tests carried out with the platforms.

Configuration Parameter	YetiMote	MarmotE SDR	B200mini and B200 USRPs	MIGOU BB Mode	MIGOU I/Q Mode
Active Clock	48 MHz	10 MHz	32 MHz	50 MHz	50 MHz
Standby Clock	32 kHz	32 kHz	Unknown	1 MHz	1 MHz
Operating Band	2.4 GHz	2.4 GHz	2.4 GHz	2.4 GHz	2.4 GHz
Modulation	GMSK	GMSK	GMSK	BFSK	GMSK
Data Rate	250 kbps	250 kbps	250 kbps	200 kbps	250 kbps
Transmission Power	0 dBm	0 dBm	0 dBm	0 dBm	0 dBm

**Table 3 sensors-19-04983-t003:** Hardware features of the different platforms evaluated.

Hardware Feature	YetiMote [16]	MarmotE SDR [29]	B200mini/B210 USRPs [18]	MIGOU
Processor	ARM Cortex-M4 Up to 80 MHz Floating Point Unit	ARM Cortex-M3 Up to 100 MHz	-	ARM Cortex-M3 Up to 142 MHz
Programmable Logic Elements	-	6 k	75 k/150 k	56k
RF PHY Transceiver	SPIRIT1 (×2) CC2500 (×1)	-	-	AT86RF215 (×1)
Operating Freq. Range	433 MHz, 868 MHz and 2.4 GHz bands	2.4 GHz band	70 MHz to 6 GHz	433 MHz, 868 MHz and 2.4 GHz bands
Access to I/Q Signals—Max. Sample Rate	-	22 MSps	61.44 MSps	4 MSps

**Table 4 sensors-19-04983-t004:** Average, maximum, and minimum power consumption of MIGOU in the evaluated modes. The values were calculated using 2000 samples acquired at 1 kHz (window of 2 s).

MIGOU Mode	Operation Mode	Power Consumption (mW)		
		Average	Maximum	Minimum
BB	Transmit	163.4	203.0	135.2
BB	Receive	154.5	190.4	133.5
I/Q	Transmit	299.6	324.4	296.7
I/Q	Receive	201.4	212.7	198.5
-	Sleep	22.8	26.8	18.8

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
