# Peer review of "MIGOU: A Low-Power Experimental Platform with Programmable Logic Resources and Software-Defined Radio Capabilities"

_sensors, 2019, doi:10.3390/s19224983_

Round 1

Reviewer 1 Report

The paper is well structured and well written, its timeliness is high, as the proliferation of SDR-based boards steadily increases. The need for devices with low-energy consumption is of highest importance. The authors have compared their platform with other popular solutions, and based on the achieved results drawn some conclusions related to the analyzed board. 

The following weaknesses could be improved:

Although some diagrams for MIGOU are provided, the board itself is the clue of the work, so I would expect more diagrams of various kinds, more technical data, more scientific analysis of it from the perspective of SDR applications It is not really clear how exactly all the power consumption values have been collected for all platforms. I would expect a detailed description of the measurement process, how it was done, if based on datasheets - also some more details on it should be provided. Also, these values provided in fig. 4 - these are single values, I believe these are some average/max values, thus I would expect some time-dependent curves (at least for MIGOU and maybe other platform) to see the real-time differences.  Please improve "This same" l308. or "whose consumption" l. 57 (power consumption I believe) 

Reviewer 2 Report

The paper "MIGOU: A Low-Power Experimental Platform with Programmable Logic Resources and Software- Defined Radio Capabilities" presents an attempt to build an experimental hardware platform named MIGOU, suitable for implementing algorithms for highly efficient and reliable communications in complex IoT environments, e.g. based on Cognitive Radio approaches.

The paper starts with motivation of the work. The authors analyze the opportunity of  developing wireless capable systems able to sense and analyze the signal spectrum, and to adapt their communication parameters for best efficiency. The authors debate on the weaknesses of current SDR platforms, used to develop such algorithms, which implements heavy computational power and therefore are not suitable to play role of a cheap battery powered IoT device. Moreover, the existing wireless end-devices, such as TelosB, MICAz  or YetiMote have no hardware resources suitable for implementation of Cognitive Radio approaches. All these aspects are then presented in detail in the Related Works section, where several SDR platforms are reviewed and their weaknesses regarding the proposed platform aim reveled.  

A detailed description of the proposed platform is presented in Section 3. According to the authors, it was designed having in mind several aspects as hybrid radio capabilities, low-powered programmable logic, and flexible hardware-software boundary.  It should be suitable for agile application development of new algorithms and should integrate power consumption metering tools. The first aspect is achieved using Atmel AT86RF215, which allows the platform to operate as a traditional node or as an SDR system. The control unit is based on SmartFusion2, which integrates an ARM Cortex-M3 processor and a FPGA. It supports a very useful low power mode named Flash*Freeze. The platform could be powered by a  single cell Li-Ion battery or using a power adapter for development purpose. As the platform OS the authors choose YetiOS, which is a rich features FreeRTOS implementation.

In order to validate the work, the authors compared their platform with some representative platforms as YetiMote - based on STM32L476RE microcontroller and three COTS radio transceivers, MarmotE SDR - a popular battery powered SDR system , B200mini and B210 USRPs - some widely used SDR platforms. Besides comparison of implemented features, the criteria covers the power consumption in transmit, receive and sleep modes. The authors conclude that they achieved state of the art trade-off between hardware flexibility and energy efficiency for an experimental platform.  

The paper is well written and organized. The work is presented in details. The references are carefully selected to motivate the approach. However, I have some suggestion for the authors:

- Please replace the word "prevents" in the sentence [66] " However, this system architecture prevents CR and edge computing research in areas of great interest, such as IoT, Wireless Sensor Networks (WSNs) or mobile sensing, or at least from doing it without the current architectural constraints.", with something like "does not supports", which express better the reality.

- It would be nice to add some tests on how this platform performs for a real mixed CR/COTS transceiver application. In my opinion the versatility and utility of a platform could be better demonstrated by testing a full application instead of just comparing row data and basic features.
